# *Cinnamomum cassia* Alleviates Neuropsychiatric Lupus in a Murine Experimental Model

**DOI:** 10.3390/nu17111820

**Published:** 2025-05-27

**Authors:** Georges Maalouly, Youakim Saliba, Joelle Hajal, Anna Zein-El-Din, Luana Fakhoury, Rouaa Najem, Viviane Smayra, Hussein Nassereddine, Nassim Fares

**Affiliations:** 1Laboratory of Research in Physiology and Pathophysiology, Faculty of Medicine, Saint Joseph University of Beirut, Beirut 1104 2020, Lebanon; georges.maalouly@usj.edu.lb (G.M.); youakim.saliba@usj.edu.lb (Y.S.); joelle.hajal2@usj.edu.lb (J.H.); annazeineldin@gmail.com (A.Z.-E.-D.); luana.fakhoury@net.usj.edu.lb (L.F.); rouaa.najem@net.usj.edu.lb (R.N.); 2Faculty of Medicine, Saint Joseph University of Beirut, Beirut 1104 2020, Lebanon; viviane.traksmayra@usj.edu.lb (V.S.); hussein.nassereddine@usj.edu.lb (H.N.)

**Keywords:** *Cinnamomum cassia*, neuropsychiatric lupus, TLR7, blood–brain barrier, neuroinflammation

## Abstract

**Background:** The pathogenesis of neuropsychiatric lupus erythematosus (NPSLE) is very complex and is associated with neuroinflammation and blood–brain barrier compromise. Experimental investigations of NPSLE have classically relied on spontaneous models. Recently, TLR7 agonist-induced lupus has been shown to exhibit similar neuropsychiatric manifestations to spontaneous ones. Cinnamon is a widespread spice and natural flavoring agent. It has been proven to modulate vascular endothelial tight junctions, neuroinflammation, and autoimmunity pathways, but it has never been tested in relation to lupus. **Hypothesis/Purpose:** In this pilot study, we aimed to explore the disease-modifying effect of *Cinnamomum cassia* on NPSLE in a TLR7 agonist-induced model. **Study Design:** An experimental design was followed in this study. **Methods:** Lupus was induced in C57BL/6J female mice via the direct application of imiquimod, a TLR7 agonist (5% imiquimod cream, 1.25 mg three times weekly), to the skin. Mice were divided into five groups (*n* = 8 per group): a sham group (S), a sham group supplemented with cinnamon (SC), an imiquimod-treated group (L), an imiquimod-treated group supplemented with cinnamon starting from induction (LC), and an imiquimod-treated group supplemented with cinnamon beginning two weeks prior to induction (CLC). This protocol was followed for six consecutive weeks. *Cinnamomum cassia* powder was administered orally at 200 mg/kg, 5 days per week. **Results:** Behavioral alterations were significantly ameliorated in the CLC group compared to lupus mice. Neuronal shrinkage and nuclear chromatin condensation were visible in the hippocampal cornu ammonis and dentate gyrus zones of lupus mice, with an increased expression of TLR7 and NLRP3, versus significantly less neurodegeneration and TLR7 and NLRP3 expression in the CLC group. In addition, the expression of the blood–brain barrier endothelial cell tight junction proteins claudin-1, occludin, and ZO-1 was abnormally modified in lupus mice and was restored in the CLC group. Moreover, while the cell–cell border delocalization of claudin-1 was documented in cultured blood–brain barrier endothelial cells treated with the plasma of lupus mice to a punctate intracytoplasmic fluorescence pattern, only cells treated with the plasma of the CLC group exhibited a complete reversal of this redistribution of claudin-1. Finally, cinnamaldehyde seemed to interact with TLR7 at multiple sites. **Conclusions:**
*Cinnamomum cassia* seems to alleviate the pathogenesis of NPSLE. Supplementation with *Cinnamomum cassia* could be of great interest to modulate the activity and severity of the disease.

## 1. Introduction

Systemic lupus erythematosus is a highly complex autoimmune disease with a female predilection that causes multiorgan damage [1]. NPSLE designates the neurological and psychiatric disorders occurring during lupus and encompasses a spectrum of phenotypes, including cognitive impairment and neuropsychiatric signs [2]. Several limitations impede the understanding of the pathophysiology of NPSLE, including the difficulty of brain tissue sampling in humans and the variability in neuropsychiatric manifestations. Mouse models of NPSLE are thus valuable tools for investigating this field [3]. Spontaneous mouse models of lupus are the most frequently used in the study of neurological dysfunction associated with this condition [4]. The imiquimod-induced lupus model was introduced in 2014 as a new model emphasizing the role of environmental modifiers in the initiation of this disease [5]. Several subsequent studies used this model successfully in female mice because of their increased sensitivity to TLR7-driven autoimmune reactions [6]. However, the neuropsychiatric dysfunction in this model is still not well-investigated. Recent studies support the interest in lupus induced in wild strains for the study of the disease’s neuropsychiatric aspects, and new data show that female mice in this model have the same neurocognitive dysfunction as MRL/lpr mice [7].

The blood–brain barrier (BBB) plays a central role in the pathogenesis of NPSLE. The BBB constitutes a selective boundary between the brain and the blood, protecting the central nervous system against the infiltration of deleterious compounds [8]. The BBB depends on junctional complexes (especially tight and adherens junctions) connecting cerebral endothelial cells. Tight junction dysfunction increases BBB permeability and results in the penetration of proinflammatory cytokines and cellular components, leading to brain injury [9]. BBB integrity is affected by inflammatory cytokines. Recent data point to a possible role of intestinal barrier dysfunction in BBB dysfunction and neuroinflammation in different neurological diseases. In addition, the structure of tight junctions is recognized to be affected by oxidative stress [10] and is targeted by antioxidants with neuroprotective properties [11].

*Cinnamomum cassia* is an aromatic tree of the Lauraceae family. Its bark is the source of cinnamon, a popular spice, and is rich in phytochemicals, especially cinnamaldehyde [12]. Cinnamon has been proven to modulate intestinal barrier function [13], neuroinflammation [14], and many autoimmunity pathways [15], particularly TLR2 and TLR4 [16], but it has never been tested in TLR7-induced diseases, including in an experimental lupus model. Therefore, in this pilot study, we aimed to explore the disease-modifying potential of cinnamon in NPSLE using a murine experimental model.

## 2. Methods

### 2.1. Animals and Experimental Protocol

This study was approved by the Ethics Committee of Saint-Joseph University of Beirut (CEHDF 1762). It adhered to the Guiding Principles for the Care and Use of Animals established by the American Physiological Society and European Parliament Directive 2010/63 EU on the ethical treatment of animals. The animals were maintained at constant temperature and humidity, provided with standard rodent chow, granted unrestricted access to tap water, and acclimated to these conditions prior to this study’s initiation.

C57BL/6J adult female mice were distributed randomly into five groups (*n* = 8 per group): a sham group (S), a sham group supplemented with cinnamon (Sham Cinna), an imiquimod-treated group (Lupus), an imiquimod-treated group supplemented with cinnamon starting from induction (Lupus Cinna), and an imiquimod-treated group supplemented with cinnamon starting two weeks before induction (Cinna Lupus Cinna). In the imiquimod-treated groups, 1.25 mg of 5% imiquimod cream was applied to the skin of the right ear of each mouse three times per week (Aldara 3M Pharmaceutical, Saint Paul, MN, USA). This protocol was followed for six weeks. Cinnamon supplementation was applied orally, 5 days per week, at 200 mg/kg (*Cinnamomum cassia*, Solgar, manufactured in the USA). Cinnamon powder was offered individually to each animal on a small, flattened piece of chow, slightly moistened with water, on a daily basis. Each time, the researcher verified that the entire piece was completely consumed within the following hour. This procedure was carefully performed during the entire experimental protocol. Note that in the text, the following abbreviations are used: S for Sham; SC for Sham Cinna; L for Lupus; LC for Lupus Cinna; and CLC for Cinna Lupus Cinna.

### 2.2. Behavioral Testing

Behavioral testing was conducted during the sixth week of the experimental design, with intervals of 24 to 48 h between consecutive tests.

Depression-like behavior was assessed using the forced swim test. After a sensitization and rest period, each animal underwent testing for 10 min, and predominant behavior was documented every 10 s. Predominant behavior was categorized as swimming, immobility (floating), and climbing [17].

Anxiety-like behavior was assessed using the elevated plus maze. Each animal was observed for 5 min, and the number of entries and time spent in the open arms was recorded as previously described [18].

A Y-maze apparatus was used to assess working memory and exploratory activity. Each mouse was placed in the central area and observed for 10 min. The number of correct alterations/total number of novel arms was calculated for each mouse as previously described [19].

### 2.3. Histological Assessment

At the time of sacrifice, the animals were anesthetized using a combination of ketamine (75 mg/kg; Interchemie, Venray, Holland) and xylazine (10 mg/kg; RotexMedica, Hamburg, Germany). Deep anesthesia was confirmed by the absence of a response to toe pinching, after which the animals were euthanized for tissue collection. Blood samples were drawn into EDTA tubes, and plasma was separated by centrifugation at 4500 rpm for 15 min and then stored at –80 °C for subsequent neural cell culture treatments. For brain extraction, the scalp was surgically removed to expose the skull. The cranial bone was cut along the sutures, and segments of the skull were excised using a scalpel blade. The connective tissue was carefully removed, and the brain was extracted from the cranial cavity. One half of the brain was preserved in 10% neutral buffered formalin for histological analysis, while the other half was used to isolate the hippocampus, which was stored at –80 °C for subsequent protein extraction. Paraffin-embedded sections of 5 μm tissues were stained with hematoxylin–eosin (HE) (Sigma-Aldrich, St. Louis, MO, USA). A semi-quantitative scoring system was used by two blinded independent pathologists. Glomerular lesions were characterized for endocapillary proliferation (0 to 2 or 0 to 3), mesangial matrix expansion and segmental sclerosis (0 = < 10%; 1 = 10–50%; 2 = > 50% of the glomeruli) [5], and for mesangioproliferation and the presence of neutrophil infiltration (0 = < 5%; 1 = 5–25%; 2 = 25–50%; 3 = > 50% of the glomeruli). Moreover, for each mouse, global glomerular lesion scores were calculated with at least 50 glomeruli.

To detect IgG accumulation in the kidneys and brain, renal sections were incubated with the goat anti-mouse IgG H&L (Alexa Fluor^®^ 594) (ab150116, Abcam, Cambridge, UK) secondary antibody at a 1/250 dilution for 30 min at 37 degrees, whereas cerebral sections were incubated with goat anti-mouse IgG H&L (Alexa Fluor^®^ 647) (ab150115, Abcam, Cambridge, UK). Pictures were taken using an Axioskop 2 immunofluorescence microscope (Carl Zeiss Microscopy GmbH, Jena, Germany) equipped with a CoolCube 1 CCD camera (MetaSystems, Newton, MA, USA).

### 2.4. Western Blotting

Kidney, brain, and intestine tissues were homogenized and lysed in an assay lysis buffer containing NaCl (150 mM), Tris-OH pH 7.5 (50 mM), EDTA (95 mM), and Triton X-100 (0.5%), along with protease and phosphatase inhibitors to extract total proteins. Protein concentrations were measured using the Bradford protein assay (Bio-Rad, Marnes-la-Coquette, France). Subsequently, the samples were denatured in Laemmli loading buffer (Bio-Rad) with 10% β-mercaptoethanol (Sigma-Aldrich) at 37 °C for 20 min. Proteins were separated on SDS 12% PAGE and transferred onto polyvinylidene fluoride membranes (Bio-Rad). The membranes were then blocked using TBS–Tween blocking solution containing 5% BSA and incubated overnight at 4 °C with various primary antibodies: tight junction protein (ZO-1) (1/1500; ab96587; Abcam, Cambridge, UK); claudin 1 (1/500; ab19098; Abcam); occludin (1/50000; ab167161; Abcam); TLR7 (1/100; ab24184; Abcam); glyceraldehyde-3-phosphate dehydrogenase (GAPDH) (1/2500; D6H11; Cell Signaling Technology, Danvers, MA, USA); T cell immunoglobulin or Kidney Injury Molecule-1 (KIM-1) (1/500; ab47634; Abcam); neutrophil gelatinase-associated lipocalin (NGAL) (1/500; ab63929; Abcam); NLRP3 (1/1000; NBP2-12446; Novus Biologicals, Centennial, CO, USA); and β-actin (1/1000; SAB1305554; Sigma-Aldrich, St. Louis, MO, USA). Following washes with TBS–Tween, the membranes were incubated for one hour at room temperature with secondary antibodies, goat anti-rabbit IgG (H+L) HRP-conjugated (Cat# 170-6515, Bio-Rad) and goat anti-mouse IgG (H+L) HRP-conjugated (Cat# 170-6516, Bio-Rad), both at a dilution of 1/3000 in blocking solution. Western blots were visualized through enhanced chemiluminescence using Clarity Western ECL Substrate (Cat# 1705060; Bio-Rad). Signal detection was carried out using an imaging system with a CCD camera (Omega Lum G, Aplegen, Gel Company, San Francisco, CA, USA), quantifications—using LICOR Image Studio Lite version 5.2. Three to six Western blots were prepared for each condition.

### 2.5. Immunoprecipitation

Proteins (100 µg) extracted from brain tissues were incubated overnight at 4 °C with a phosphotyrosine monoclonal antibody (2 μg; CSB-MA080265; Cusabio, Houston, TX, USA). This was followed by incubation with prewashed protein A Sepharose beads (50 μL; ab193256, Abcam, Cambridge, UK) at 4 °C for 2.5 h. After several washes with a lysis buffer, the proteins were eluted with 20 μL of 2× Laemmli loading buffer (1610737EDU; Bio-Rad Laboratories Inc., Irvine, CA, USA) containing β-mercaptoethanol in addition to 20 μL of glycine (pH 2.5). Eluted proteins were separated via SDS 10% PAGE, blotted on a PVDF membrane, and incubated with the ZO-1, claudin 1, and occludin antibodies (Abcam, Cambridge, UK). The VeriBlot for IP Detection Reagent (HRP) (ab131366; Abcam, Cambridge, UK) was operated to detect immunoprecipitated proteins while avoiding interference from IgG heavy (50 kDa) and light chains (25 kDa). Visualization was carried out using enhanced chemiluminescence (Clarity Western ECL, Biorad, CA, USA).

### 2.6. Statistical Analysis

Statistical analysis was conducted using GraphPad Prism 9, with data expressed as the means ± SEM. Sample sizes (N) are detailed in each figure legend. The Shapiro–Wilk normality test was employed to determine whether the populations adhered to a Gaussian distribution. For comparisons involving more than two groups, ordinary one-way ANOVA was applied when Gaussian distribution criteria were met, followed by Sidak’s post hoc test to identify specific group differences contributing to ANOVA significance. In cases of non-normal data distribution, Kruskal–Wallis ANOVA on ranks was utilized, followed by Dunn’s multiple comparisons test.

## 3. Results

### 3.1. Cinnamon Supplementation Attenuates Lupus Nephritis

Significant increases in mesangial proliferation, endocapillary proliferation, and segmental glomerulosclerosis were observed in the lupus group, with increased neutrophil infiltration and inflammation scores (Figure 1A,B), as well as prominent glomerular IgG deposition (Figure 1D). However, these histological features were alleviated in the CLC group and were associated with less IgG deposition (Figure 1D). Moreover, cinnamon-supplemented groups (LC and CLC) showed lower kidney protein expression of KIM-1 and NGAL (Figure 1C). No significant differences were noted between the S and SC groups, nor between LC and CLC, although a tendency towards less damage was observed in the CLC group.

### 3.2. Cinnamon Supplementation Modulates Intestinal Tight Junction Proteins in Lupus

Claudin, occludin, and ZO-1 constitute the primary proteins of intestinal tight junctions. Our results showed a marked decrease in these proteins in the lupus groups. Nevertheless, this decrease was significantly less prominent in cinnamon-treated mice, particularly for claudin and ZO-1 (Figure 2). No significant differences were found between the S and SC groups, nor between LC and CLC.

### 3.3. Cinnamon Ameliorates Brain Histopathology and Inflammatory Markers in Lupus

Nuclear chromatin condensation and neuronal shrinkage were observed in the hippocampal cornu ammonis and dentate gyrus zones in lupus mice, with no obvious leukocyte infiltration (Figure 3A,B). Semi-quantitative scoring showed less degeneration in the CLC group (Figure 3C). In addition, TLR7 and NLRP3 expression was significantly increased in the lupus groups, although it was significantly less pronounced in the CLC group (Figure 3D). Moreover, hippocampal IgG deposition was confirmed in the lupus groups and was less marked in the CLC group (Figure 3E). No significant differences were found between the S and SC groups, nor between LC and CLC.

### 3.4. Cinnamon Improves Behavioral Impairment in Lupus

In the elevated plus maze test, the frequency of entries, as well as the time in the open arms, were significantly higher in the CLC group in comparison to the untreated lupus mice (Figure 4A,B). The percent alteration in the Y-maze test was markedly reduced in the lupus mice compared to the CLC group (Figure 4C). Immobility behavior frequency in the forced swim test was significantly increased in the lupus groups (Figure 4D), although this increase was significantly less pronounced in the CLC group (Figure 4D). No significant differences were found between the S and SC groups, nor between LC and CLC, in all the tests.

### 3.5. Cinnamon Modulates Brain Tight Junctions in Lupus

Total claudin-1, occludin, and ZO-1 expression was significantly upregulated in the lupus mice and restored in the CLC group (Figure 5). Furthermore, the expression of Tyr-phosphorylated forms of claudin-1, occludin, and ZO-1 was increased in the lupus mice. However, this increase was seen to be reversed only for the phosphorylated forms of claudin-1 and occludin in the CLC group (Figure 5). No significant differences were found between the S and SC groups, nor between LC and CLC.

Moreover, while cell–cell border delocalization of claudin-1 was documented in the BBB endothelial cells treated with the plasma of the lupus mice as a punctate intracytoplasmic fluorescence pattern, only the cells treated with the plasma of the CLC group exhibited a complete reversal of this redistribution of claudin-1 (Figure 6).

### 3.6. Cinnamaldehyde Interacts with TLR7

Our molecular docking study revealed that cinnamaldehyde, a principal constituent of cinnamon, interacts with TLR7 in multiple sites and may impede TLR7 activation by imiquimod. Imiquimod binds to site 1 in the TLR7 dimerization interface and activates the TLR7 dimer. Imiquimod interacts with TLR7 via three hydrogen bonds at cysteine 189 (CYS189), tyrosine 190 (TYR190), and serine 192 (SER192), along with a covalent bond at asparagine 215 (ASN215). When the TLR7/imiquimod couple was docked with cinnamaldehyde, the latter seemed to lodge in the same pocket of the TLR7 homodimer, interacting with ASN215 via a hydrogen bond. Three other non-covalent interactions were also present at proline 261 (PRO261), CYS270, and TYR468.A cellular thermal shift assay was performed to validate the molecular docking predictions of cinnamon’s interaction with TLR7. Cinnamon demonstrated a modulating effect on TLR7 stability in neuronal cells (Appendix A). CETSA results revealed that TLR7 exhibited enhanced thermal stability in the presence of cinnamaldehyde compared to cells treated with vehicle alone, as indicated by a higher protein expression under heat conditions. Imiquimod was used as a positive control and enhanced TLR7 thermal stability as expected (Appendix A).

## 4. Discussion

To the best of our knowledge, this study is the first to demonstrate that oral cinnamon supplementation improves NPSLE in an induced model using wild-type mice. The beneficial effects of cinnamon were especially evidenced in the preventive arm, as documented by behavioral tests, brain histology and inflammatory parameters, and brain tight junction protein expression. While spontaneous murine lupus models focus mainly on the genetic basis of lupus, induced models of lupus in wild murine strains highlight the presumed pathways by which nutrition and external factors contribute to the development of lupus; however, their use in the study of neurological dysfunction in this disease has been scarce. To explore the differential impact of the timing of cinnamon administration, we evaluated two treatment arms: cinnamon supplementation initiated two weeks prior to induction and supplementation started at the time of induction. TLR7 activation by imiquimod, a TLR7 agonist, was the cornerstone of experimental lupus in our study. The epicutaneous application of imiquimod activated plasmacytoid dendritic cells through the systemic release of interferon and the subsequent development of autoimmunity [5] and gut permeability dysfunction [20]. The present study reproduces the finding that wild-type mice exposed to the topical application of imiquimod three times weekly manifest a systemic lupus-like disease [5]. Histological glomerulonephritis lesions, along with an increase in kidney injury markers KIM-1 and NGAL, were observed in the imiquimod groups within our model, validating lupus induction in the imiquimod-treated mice. While this was beyond the scope of our study objectives, cinnamon treatment, especially in the CLC group, seemed to improve histopathology and kidney injury markers related to lupus nephritis. These findings indicate promising directions for future research into the nephroprotective properties of cinnamon in lupus pathology.

Data on neurological dysfunction in the imiquimod-induced lupus model are sparse. Recently, imiquimod-induced lupus mice were found to have cognitive impairment, brain injury, and heightened expression of hippocampal microglia CD40 similar to those of MRL/lpr mice [7]. Our study confirms brain histopathological lesions in this model and adds new elements to the description of depressive and anxiety features besides cognitive dysfunction, with an ameliorative effect of preventive cinnamon treatment. Interestingly, an increased expression of TLR7 in hippocampal cells was documented in our study. This may be related to the systemic release of interferon following plasmacytoid dendritic cell activation by imiquimod and the induction of TLR transcription via interferon-stimulated response elements [5,21,22,23].

Cinnamon exhibits several antioxidant and immunomodulating properties by modifying many pathways, such as TLR2 and TLR4, COX-2, iNOS, p38-MAPK, NF-κB, and others [15]. In experimental autoimmune encephalitis, cinnamon has the potential to inhibit T cell migration via the BBB and intrathecal T cell proliferation [14]. In addition, cinnamaldehyde treatment ameliorates chronic stress-induced depressive-like behaviors and neuroinflammation in middle-aged rats [24]. However, to our knowledge, the interaction of cinnamon with TLR7-dependent pathophysiological pathways has not yet been elucidated, and cinnamon has never been tested as a disease modifier in lupus.

In our model of TLR7 agonist-induced lupus, preventive supplementation with cinnamon seemed to alleviate intestinal alterations of TJ expression. This finding is aligned with previous studies showing that cinnamaldehyde’s positive effect on intestinal barrier function is associated with an increased abundance of claudin-4, ZO-1, ZO-2, and ZO-3 and the promotion of claudin-1 and claudin-3 localization to the plasma membrane on immunofluorescence staining [25]. We previously showed that imiquimod-induced lupus is associated with altered intestinal permeability, increased fecal calprotectin, decreased TJ expression, and liver bacterial translocation [20]. This could be driven by TLR7 and interferon [20,26,27]. Intestinal barrier dysfunction can trigger immune cell activation, compromising the blood–brain barrier and facilitating neuroinflammation and neuronal damage in some neurodegenerative diseases [28]. Cinnamon’s ameliorative effect on intestinal TJs may thus be one of the pathways for modulating the neuropsychiatric manifestations of lupus.

In our experimental model, cinnamon’s amelioration of neurologic dysfunction, documented by histology and behavioral tests, was paralleled by a reduction in TLR7 and NLRP3 expression in the brain. Pretreatment with cinnamon two weeks before induction, followed by continuous treatment during the induction phase, was the most beneficial in terms of TLR7 and NLRP3 downregulation, neurodegeneration, and behavioral impairment, in comparison with treatment started concomitantly with induction. Thus, the neuroprotective effect of cinnamon was more efficient when mice were supplemented sub-chronically before exposure to the environmental activator of the TLR7 pathway.

Moreover, alterations of brain TJ expression were also reversed by the preventive supplementation of cinnamon. TJ expression and cellular traffic in the brain are complex. Brain microvascular endothelial cells in mice express claudin-5, −12, and − 25 in normal physiology and claudin-1 under modified conditions [29]. Claudin-1 mRNA and protein expression were upregulated in association with BBB leakage. This was associated with the chronic phase of murine stroke and with an endothelial proinflammatory phenotype [30]. Phosphorylation of TJs on different residues affects their cellular traffic and seems to be tissue-specific. The phosphorylation of specific residues in occludin and ZO-1 is essential for maintaining the barrier; however, further phosphorylation on other residues, particularly tyrosine, results in barrier compromise [29]. In rat brain endothelium, hypoxia is associated with an enhanced expression of ZO-1, occludin, and claudin-5 [31]. This is in contrast to other authors’ results [32] demonstrating that IL-17 disrupts the BBB and results in a decreased expression of occludin and ZO-1. In another study, no significant changes in the expression of claudin-5, ZO-1, or occludin were retrieved; however, an enhanced serine/threonine phosphorylation of ZO-1 in response to IL-1β was demonstrated [33]. Tyrosine phosphorylation of claudin-5 contributes to enhanced paracellular permeability in brain endothelial cells, accompanied by mononuclear cell infiltration through the compromised BBB [29]. Our study shows that the plasma of CLC mice does not disrupt endothelial cell TJ distribution compared with the plasma of L and LC mice. Altogether, these results suggest that preventive cinnamon supplementation protects the BBB from disruption in TLR7-induced lupus; this protection may also mediate cinnamon’s positive effect on behavior and brain histology.

Finally, molecular docking technology showed that cinnamaldehyde, which is an essential compound of cinnamon, interacts with TLR7 at many sites and may decrease the interaction between imiquimod and TLR7 (Appendix A). While outside the scope of our article, the thermal shift assay, a technique that enables the simple determination of protein stability and ligand interactions [34], suggests that the presence of cinnamon as a whole compound modulates the stability of the imiquimod–TLR7 complex (Appendix A). These preliminary data may open novel directions in the study of cinnamon’s bioactive compounds as modulators of TLR7.

Taken together, our results suggest that cinnamon’s protective effect is, at least partly, dependent on the TLR7 signaling pathway.

## 5. Conclusions

*Cinnamomum cassia* seems to alleviate the pathogenesis of NPSLE. Our pilot study paves the way for the use of phytochemicals as an early preventive nutritional measure to cool down the inflammatory milieu before the induction of autoimmunity by environmental triggers.

## Figures and Tables

**Figure 1 nutrients-17-01820-f001:**
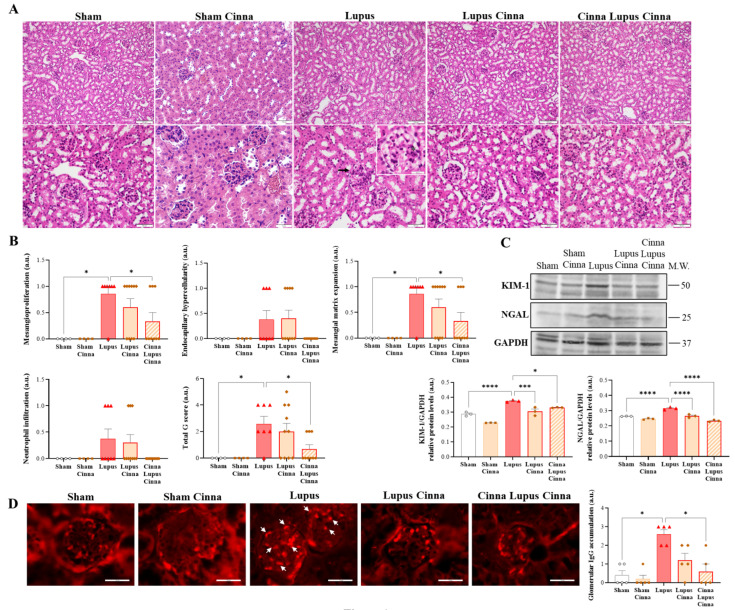
Cinnamon supplementation attenuates renal damage in experimental systemic lupus erythematosus. (**A**) Representative images of renal histological sections from different animal groups stained with hematoxylin–eosin. Arrows in the lupus group show endocapillary hypercellularity. In the upper panels, scale bars are 100 μm, and magnification is ×200. In the bottom panels, scale bars are 50 μm, and magnification is ×400. (**B**) Quantification of different renal damage scores. (**C**) Western blots and quantifications of KIM-1 and NGAL in the kidneys of the different animal groups, with GAPDH as an internal control (n = 3 for each condition). (**D**) Representative images and quantification of immunofluorescence staining (using goat anti-mouse IgG H&L Alexa Fluor^®^ 594, Ex: 590 nm; Em: 617 nm) for glomerular IgG. Arrows in the lupus group show IgG glomerular accumulation. Scale bars: 25 μm; magnification: ×400. Cinna: cinnamon; a.u.: arbitrary units. Note: * *p* < 0.05, *** *p* < 0.001, and **** *p* < 0.0001.

**Figure 2 nutrients-17-01820-f002:**
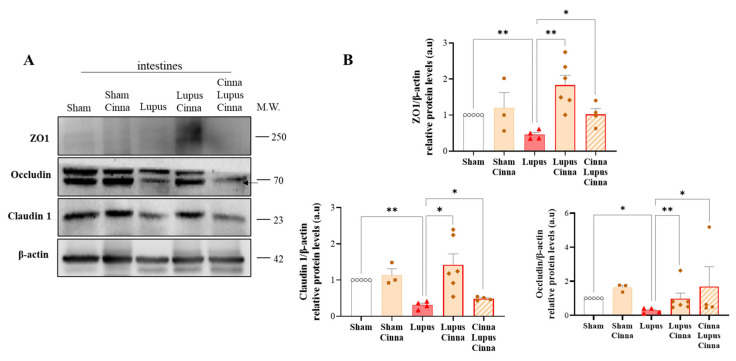
Cinnamon supplementation modulates intestinal tight junction protein expression in lupus mice. (**A**–**B**) Western blots and quantifications of the tight junction proteins claudin 1 (**A**), occludin (**B**), and ZO-1 (**B**) in the intestines of the different animal groups, with β-actin as an internal control (n = 3 for each condition). The arrow in A indicates the specific occludin band. Cinna: cinnamon; a.u.: arbitrary units. Note: * *p* < 0.05, ** *p* < 0.01.

**Figure 3 nutrients-17-01820-f003:**
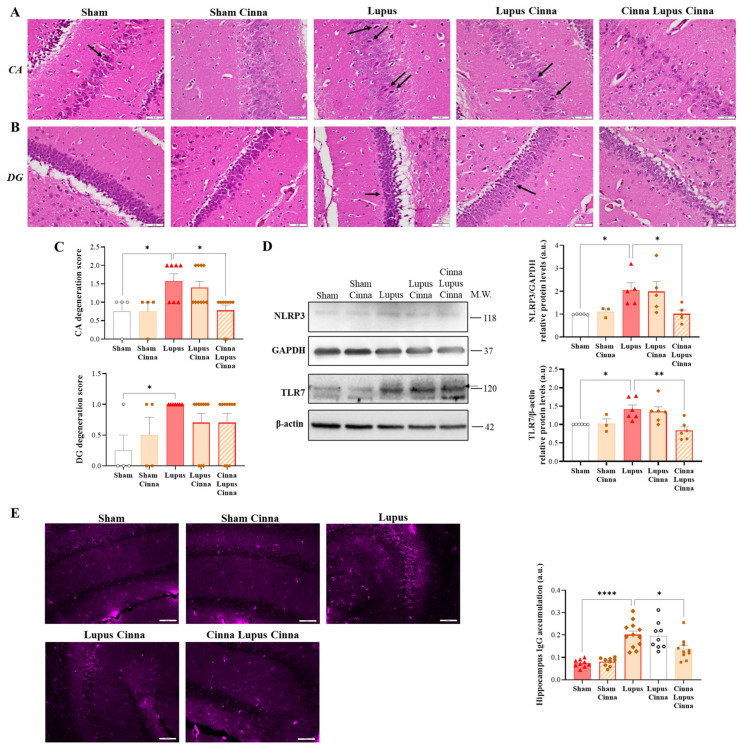
Neuroinflammation is ameliorated in lupus mice treated with cinnamon. (**A**,**B**) Representative images of hippocampal histological sections from different animal groups stained with hematoxylin–eosin. Arrows in the lupus group show neuronal degeneration. The upper panels (**A**) show the cornu ammonis zone, and the bottom panels (**B**) show the dentate gyrus zone. Scale bars: 50 μm; magnification: ×400. (**C**) Quantification of neuronal degeneration. (**D**) Western blots and quantifications of NLRP3 and TLR7 in the hippocampus of the different animal groups, with GAPDH and β-actin as internal controls (n = 3 for each condition). The arrow in D indicates the specific TLR7 band. (**E**) Representative images and the quantification of immunofluorescence staining (using goat anti-mouse IgG H&L Alexa Fluor^®^ 647, Ex: 650 nm, Em: 665 nm) for hippocampal IgG. Scale bars: 50 μm; magnification: ×400. Cinna: cinnamon; a.u.: arbitrary units. Note: * *p* < 0.05, ** *p* < 0.01, and **** *p* < 0.0001.

**Figure 4 nutrients-17-01820-f004:**
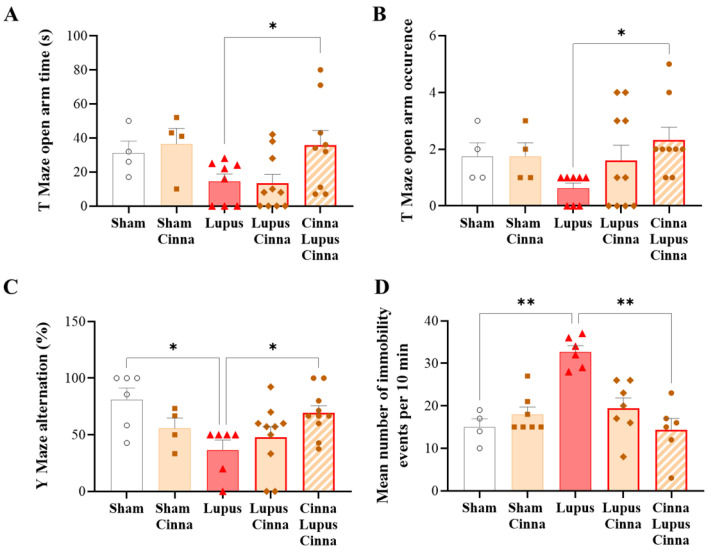
Behavioral impairment ameliorated in the lupus mice treated with cinnamon. (**A**–**D**) Behavioral tests to assess depressive-like behavior (forced swim test) and spatial working and exploratory memory (Y- and T-maze tests) in the different animal groups. Cinna: cinnamon. Note: * *p* < 0.05 and ** *p* < 0.01.

**Figure 5 nutrients-17-01820-f005:**
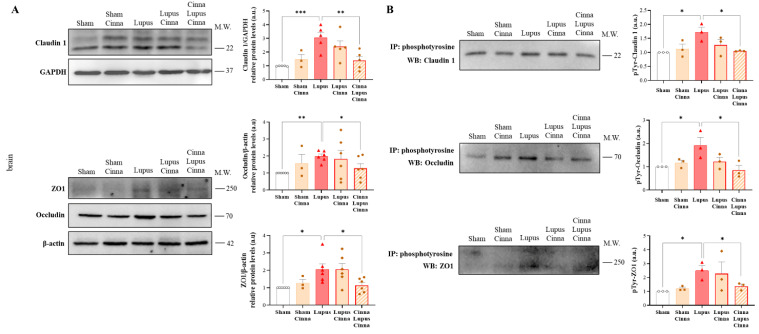
Cinnamon treatment modulates cerebral tight junction phosphorylation in lupus mice. (**A**) Western blots and quantifications of the tight junction proteins claudin-1, occludin, and ZO-1 in the brains of the different animal groups, with β-actin as an internal control (n = 3 for each condition). (**B**) Western blots and quantifications of the phosphorylated tight junction proteins claudin-1, occludin, and ZO-1 in the brains of the different animal groups, with β-actin as an internal control (n = 3 for each condition). Immunoprecipitation was performed using a phosphotyrosine antibody. Cinna: cinnamon; a.u.: arbitrary units. Note: * *p* < 0.05, ** *p* < 0.01, and *** *p* < 0.001.

**Figure 6 nutrients-17-01820-f006:**
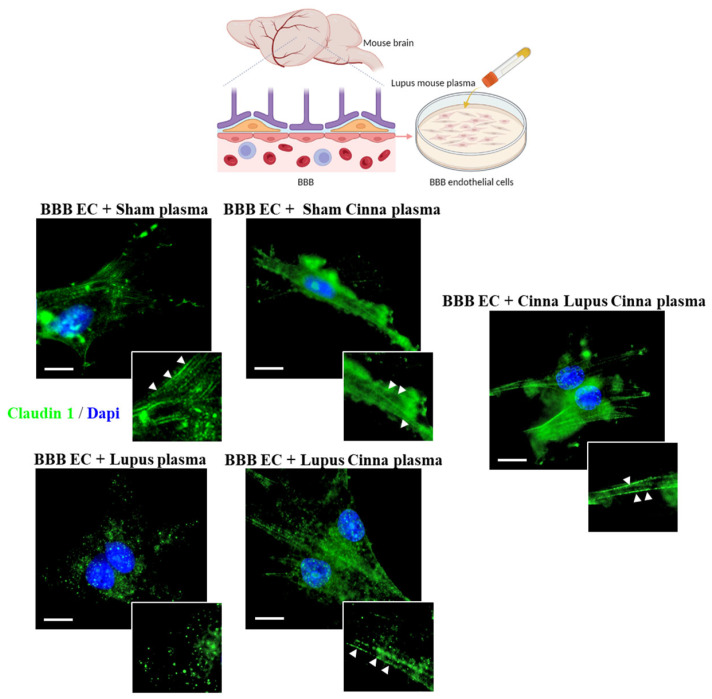
Cinnamon affects tight junction distribution in blood–brain barrier endothelial cells. Representative images of immunofluorescence staining (488 nm) for claudin-1 in blood–brain barrier (BBB) endothelial cells cultured and treated with plasma from the different animal groups. Nuclei were stained with DAPI. Arrows show the linear distribution of claudin-1 along the plasma membrane; n = 3 (immunofluorescence from three cultures). Scale bars: 20 μm; magnification: ×1000. Cinna: cinnamon.

## Data Availability

Data available upon request.

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
