# Peer review of "Cinnamomum cassia Alleviates Neuropsychiatric Lupus in a Murine Experimental Model"

_nutrients, 2025, doi:10.3390/nu17111820_

Round 1
Reviewer 1 Report
Comments and Suggestions for Authors
This is an interesting article.
Western blots for occludin and TLR7 showed (Figs. 2 and 3) two bands. Please indicate which is non-specific and which bands were used for quantification.
Please provide details on how the cinnamon was administered. Was it as an aqueous suspension? If so, what concentration was used and the volume for oral administration.
What is the concentration of the active compound (cinnamaldehyde) in the cinnamon supplement?
The authors used the abbreviation "Neuropsychiatric lupus erythematosus (NPSLE)". Please use this abbreviation throughout the manuscript. Indeed, in the Abstract we can find (lines 20–21): "We aim in this pilot study to explore the disease modifying effect of Cinnamomum cassia on NPSLE in the TLR7 agonist-induced model.” However, in the Introduction (lines 74–75) there is no abbreviation: “Therefore, in this pilot study, we aim to explore the disease-modifying potential of cinnamon in neuropsychiatric lupus in a murine experimental model.” Similarly for lines 49, 51, 188, etc.
The order of description in the Results should be from Figure A to D (see section 3.4), but not vice versa.
Check for typos: lines 48, 58, 72, 290 etc.
ZO1, but not Zo1 (see lines 248, 250) etc.
I don't have additional comments for supplemental material and ARRIVE checklist.
Author Response
We sincerely appreciate and are grateful for the constructive comments and insightful feedback provided by the reviewers and the editorial team, which has significantly improved our manuscript as requested. We have carefully addressed all concerns and revised the manuscript accordingly (additions and modifications are written in red within the revised manuscript). Please find below a detailed point-by-point response to each comment.
We hope that our revisions all concerns. It is truly an honor for us to publish our work in your esteemed journal, and we deeply appreciate the opportunity to contribute to this distinguished platform.
Thank you for your time and consideration.
Sincerely,
Nassim FARES
On behalf of all authors
Reviewer 1 :
- Comments 1: Western blots for occludin and TLR7 showed (Figs. 2 and 3) two bands. Please indicate which is non-specific and which bands were used for quantification.
Response 1: The bands used for quantification of occludin and TLR7 were indicated by an arrow on the right side of the western blot photo; please refer to the figure 2 and figure 3 revised version. In addition, this was mentioned in the figures 2, 3 legends.
- Comments 2: Please provide details on how the cinnamon was administered. Was it as an aqueous suspension? If so, what concentration was used and the volume for oral administration.
Response 2: Cinnamon powder was offered daily to each animal individually on a small flattened piece of chow, which was slightly moistened with water. The researcher verified each time that the entire piece was completely consumed within the following hour. This procedure was carefully performed during the entire period of the experimental protocol.
- Comments 3: What is the concentration of the active compound (cinnamaldehyde) in the cinnamon supplement?
Response 3: Our study objective is to use the cinnamon bark as a nutritional neuroprotective intervention in lupus. Cinnamomum cassia was used as a whole spice (as in many other studies). This administration, rather than isolated compound of the spice, is closer to human real-life nutrition and more easily translated to nutritional modifications in clinical studies and epidemiology.
Based on the typical cinnamaldehyde content of the Cinnamomum cassia ground component in several commercial cinnamon supplements, a Solgar Cinnamon capsule (a high-quality mixture of grounded cinnamon and cinnamon extract) likely contains at least 8 mg of cinnamaldehyde.
- Comments 4: The authors used the abbreviation "Neuropsychiatric lupus erythematosus (NPSLE)". Please use this abbreviation throughout the manuscript. Indeed, in the Abstract we can find (lines 20–21): "We aim in this pilot study to explore the disease modifying effect of Cinnamomum cassia on NPSLE in the TLR7 agonist-induced model.” However, in the Introduction (lines 74–75) there is no abbreviation: “Therefore, in this pilot study, we aim to explore the disease-modifying potential of cinnamon in neuropsychiatric lupus in a murine experimental model.” Similarly, for lines 49, 51, 188, etc.
Response 4: We replaced the expression by the abbreviation according to your request in the revised manuscript.
- Comments 5: The order of description in the Results should be from Figure A to D (see section 3.4), but not vice versa.
Response 5: This was rectified as follow in the text: “In the elevated plus maze test, the frequency of entries, as well as the time duration in the open arms, were significantly higher in the CLC group in comparison to untreated lupus mice (Figure 4 A and 4B). The percent alternation in the Y maze test was markedly reduced in lupus mice compared to the CLC group (Figure 4C). Immobility behavior frequency in the forced swim test was significantly increased in lupus groups (Figure 4D). However, this increase was significantly less pronounced in the CLC group (Figure 4D).”
- Comments 6: Check for typos: lines 48, 58, 72, 290 etc.
Response 6: Text was checked for typos and corrected accordingly.
- Comments 7: ZO1, but not Zo1 (see lines 248, 250) etc.
Response 7: It was corrected as requested.
Reviewer 2 Report
Comments and Suggestions for Authors
The present manuscript demonstrated the effects of Cinnamomum cassis in TLR7 agonist-induced lupus model mice. It was interesting but the presentation had some concerns.
Abstract; "less neurodegeneration, and TLR7 and NLRP3 expression in the CLC groups", how about the changes in lupus? "cinnamaldehyde seems to interact with TLR7 in multiple sites" was the data of molecular docking shown in Supplementary; however, it was not described in Results section. They should be written in the text, not in Supplementary. The figure was put in Supplementary Figures.
Methods; the authors used female mice. Why did the authors use female? The background for pathology of systemic lupus erythematosus should be added in Introduction.
Methods; the protocol of Cinnamon supplementation, 5 days per week, what was the detail?
Methods; IgG accumulation, what were the suppliers, location, and product number of goat anti-mouse secondary antibody (anti-mouse IgG)? And, what did "594 nm" mean? The wavelength of absorbance? What did the authors use as a color substance and what was conjugated in antibody?
Methods; Western Blotting, were the antibodies only primary antibodies? Did the authors use horseradish peroxidase-conjugated primary antibodies? If not so, secondary antibodies should be written. What was the manufacture for the reagent of enhanced chemiluminescence?
In Results, did not Sham+Cinna have any changes? It should be described in text.
The results of LC and CLC groups, such as in Figure 1 and 2, were different in the extent. Discussion for the differences should be added.
Figure 2A, the image of Occludin was not correct. The upper band in CLC groups was lost.
The band images in Figure 2A and Figure 3D did not seem to match the quantitative graph. More representative images should be shown. In addition, there were two bands in some proteins. How did the authors measure the levels?
Figure 3, why did TLR7 agonist increase TLR7 expression? It should be discussed.
The figure legend in Figure 3, "immunofluorescence staining (647 nm)", what was 647 nm? The wavelength was difference from it in Methods, and was it absorbance? If the measurement was fluorescence, the wavelengths for excitation and emission should be written. Or, the filter used in the microscope and the system including the names of manufacture and product should be added.
In the present experimental model, the direct skin application of imiquimod, why were the changes in the protein expressions of tight junctions apposite in intestine and brain? They should be discussed.
3.4 and Figure 4, the order of A-D and the text would be better to be replaced to match in order from A to D. Moreover, Figure 4A and 4B, did not lupus have significant difference compared to sham?
Author Response
Reviewer 2:
The present manuscript demonstrated the effects of Cinnamomum cassis in TLR7 agonist-induced lupus model mice. It was interesting but the presentation had some concerns.
- Comments 1: Abstract; "less neurodegeneration, and TLR7 and NLRP3 expression in the CLC groups", how about the changes in lupus?
Response 1: We added to the abstract as requested: “with increased expression of TLR7 and NLRP3 in lupus mice”.
- Comments 2: "cinnamaldehyde seems to interact with TLR7 in multiple sites" was the data of molecular docking shown in Supplementary; however, it was not described in Results section. They should be written in the text, not in Supplementary. The figure was put in Supplementary Figures.
Response 2: As requested data of supplementary figure related to cinnamaldehyde interaction with TLR7 was added to the text in the results section. However the related methodology and the figure remained as supplementary material.
- Comments 3: Methods; the authors used female mice. Why did the authors use female? The background for pathology of systemic lupus erythematosus should be added in Introduction.
Response 3: Background on this topic was added in the introduction as requested: “ Systemic lupus erythematosus is an autoimmune disease of high complexity and multiorgan damage with female predilection… Several subsequent studies used this model successfully in female mice because of their increased sensitivity to TLR7-driven autoimmune reactions [6]… Recently the interest of induced lupus in wild strains for the study of neuropsychiatric aspect of lupus was supported, and new data show that female mice in this model have the same neuro- cognitive dysfunction as MRL/lpr mice [7].”
- Comments 4: Methods; the protocol of Cinnamon supplementation, 5 days per week, what was the detail?
Response 4 : Cinnamon powder was offered daily to each animal individually on a small flattened piece of chow, which was slightly moistened with water. The researcher verified each time that the entire piece was completely consumed within the following hour. This procedure was carefully performed during the entire period of the experimental protocol. This was added to the text.
- Comments 5: Methods; IgG accumulation, what were the suppliers, location, and product number of goat anti-mouse secondary antibody (anti-mouse IgG)? And, what did "594 nm" mean? The wavelength of absorbance? What did the authors use as a color substance and what was conjugated in antibody?
Response 5: The lacking information was added to the text : “To detect IgG accumulation in kidneys and brain, the renal sections were incubated with the Goat Anti-Mouse IgG H&L (Alexa Fluor® 594) (ab150116, Abcam, Cambridge, UK) secondary antibody at a 1/250 dilution for 30 min at 37 degrees, whereas the cerebral sections were incubated with the Goat Anti-Mouse IgG H&L (Alexa Fluor® 647) (ab150115, Abcam, Cambridge, UK). Pictures were taken using an Axioskop 2 immunofluorescence microscope (Carl Zeiss Microscopy GmbH, Jena, Germany) equipped with a CoolCube 1 CCD camera (MetaSystems, Newton, Massachusetts, USA).”
- Comments 6: Methods; Western Blotting, were the antibodies only primary antibodies? Did the authors use horseradish peroxidase-conjugated primary antibodies? If not so, secondary antibodies should be written. What was the manufacture for the reagent of enhanced chemiluminescence?
Response 6 : This was clarified in the text as requested: “ Membranes were then blocked using TBS‐Tween blocking solution containing 5% BSA and incubated overnight at 4°C with various primary antibodies: Tight junction protein (ZO1) (1/1500; ab96587; Abcam, Cambridge, UK); claudin 1 (1/500; ab19098; Abcam); Occludin (1/50000; ab167161; Abcam); TLR7 (1/100; ab24184; Abcam); glyceraldehyde-3-phosphate dehydrogenase (GAPDH) (1/2500; D6H11; Cell Signaling Technology; MA, USA); T-cell immunoglobulin or Kidney Injury Molecule -1 (KIM-1) (1/500; ab47634; Abcam); Neutrophil gelatinase-associated lipocalin (NGAL) (1/500; ab63929; Abcam); NLRP3 (1/1000; NBP2-12446; Novus Biologicals, Centennial CO, USA); β-actin (1/1000; SAB1305554; Sigma-Aldrich, St Louis MO USA). Following washes with TBS-Tween, the membranes were incubated for one hour at room temperature with secondary antibodies: Goat anti-rabbit IgG (H+L) HRP-conjugated (Cat# 170-6515, Bio-Rad) and Goat anti-mouse IgG (H+L) HRP-conjugated (Cat# 170-6516, Bio-Rad), both at a dilution of 1/3000 in blocking solution, visualization of western blots was achieved through enhanced chemiluminescence using Clarity Western ECL Substrate (Cat# 1705060; Bio-Rad). Signal detection was carried out using an imaging system with a CCD camera (Omega Lum G, Aplegen, Gel Company, SF, USA), and quantifications were conducted using Licor Image Studio Lite version 5.2. Three to six western blots were prepared for each condition.”
- Comments 7: In Results, did not Sham+Cinna have any changes? It should be described in text. The results of LC and CLC groups, such as in Figure 1 and 2, were different in the extent. Discussion for the differences should be added.
Response 7: Descriptions were added as requested. The difference in extent between LC and CLC were not significant.
- Comments 8: Figure 2A, the image of Occludin was not correct. The upper band in CLC groups was lost.
Response 8: The bands used for quantification of occludin were indicated by an arrow on the right side of the western blot photo; please refer to the figure 2 revised version. In addition, this was mentioned in the figures 2 legend.
- Comments 9: The band images in Figure 2A and Figure 3D did not seem to match the quantitative graph. More representative images should be shown. In addition, there were two bands in some proteins. How did the authors measure the levels?
Response 9: The specific band used for quantification for figures 2 and 3 were indicated by an arrow on the right side of the western blot photo. For quantification, please check raw data in supplemental material 2 and 3.
- Comments 10: Figure 3, why did TLR7 agonist increase TLR7 expression? It should be discussed.
Response 10 : As requested we addressed this issue in the discussion. Please find below a developed explanation:
The primary mechanism by which imiquimod exerts its effects is by binding to and activating intracellular TLR7, predominantly found in immune cells such as plasmacytoid dendritic cells, B cells, and to some extent, macrophages and keratinocytes. Upon activation, TLR7 recruits the adaptor protein MyD88, initiating a signaling cascade, ultimately leading to the activation and nuclear translocation of transcription factors like NF-κB and Interferon Regulatory Factors (IRFs), particularly IRF7.
These activated transcription factors then induce the expression of a wide array of inflammatory cytokines and chemokines, notably type I interferons (IFN-α and IFN-β). Accumulating evidence from animal studies suggests that type I interferons can, in turn, upregulate the expression of various TLRs, including TLR7. This creates a positive feedback loop where imiquimod-induced TLR7 activation leads to the production of type I interferons, which then further enhance TLR7 expression, potentially amplifying the immune response.
While the precise transcriptional mechanisms by which type I interferons increase TLR7 expression are still under investigation, studies have identified interferon-stimulated response elements (ISREs) and NF-κB binding sites in the promoter region of the TLR7 gene, suggesting that both IRFs and NF-κB can directly or indirectly regulate TLR7 transcription.
References :
Li ZJ, Sohn KC, Choi DK, Shi G, Hong D, Lee HE, Whang KU, Lee YH, Im M, Lee Y, Seo YJ, Kim CD, Lee JH. Roles of TLR7 in activation of NF-κB signaling of keratinocytes by imiquimod. PLoS One. 2013 Oct 11;8(10):e77159. doi: 10.1371/journal.pone.0077159. PMID: 24146965; PMCID: PMC3795621.
Lovelock DF, Liu W, Langston SE, Liu J, Van Voorhies K, Giffin KA, Vetreno RP, Crews FT, Besheer J. The Toll-like receptor 7 agonist imiquimod increases ethanol self-administration and induces expression of Toll-like receptor related genes. Addict Biol. 2022 May;27(3):e13176. doi: 10.1111/adb.13176. PMID: 35470561; PMCID: PMC9286850.
Sirén J, Pirhonen J, Julkunen I, Matikainen S. IFN-alpha regulates TLR-dependent gene expression of IFN-alpha, IFN-beta, IL-28, and IL-29. J Immunol. 2005 Feb 15;174(4):1932-7. doi: 10.4049/jimmunol.174.4.1932. PMID: 15699120.
Comments 11: The figure legend in Figure 3, "immunofluorescence staining (647 nm)", what was 647 nm? The wavelength was difference from it in Methods, and was it absorbance? If the measurement was fluorescence, the wavelengths for excitation and emission should be written. Or, the filter used in the microscope and the system including the names of manufacture and product should be added.
Response 11: This was clarified in the figure 3 legend: “Neuroinflammation is ameliorated in lupus mice treated with cinnamon. A, B: Representative images of hippocampal histological sections stained in hematoxylin eosin obtained from different animal groups. Arrows in the lupus group show neuronal degeneration. The upper panels (A) show the cornu ammonis zone and the bottom panels (B) show the dentate gyrus zone. Scale bars: 50 μm, magnification: x400. C: Quantification of neuronal degeneration. D: Western blots and quantifications of NLRP3 and TLR7 in the hippocampus of the different animal groups, with GAPDH and β-actin as an internal control (n=3 for each condition). The arrow in D indicates the specific TLR7 band. E: Representative images and quantification of immunofluorescence staining (using Goat Anti-Mouse IgG H&L Alexa Fluor® 647, Ex: 650nm, Em: 665nm) for hippocampal IgG. Scale bars: 50 μm, magnification: x400. Cinna: cinnamon; a.u.: arbitrary units. *p<0.05, **p<0.01, and ****p<0.0001.” and in the methodology : “To detect IgG accumulation in kidneys and brain, the renal sections were incubated with the Goat Anti-Mouse IgG H&L (Alexa Fluor® 594) (ab150116, Abcam, Cambridge, UK) secondary antibody at a 1/250 dilution for 30 min at 37 degrees, whereas the cerebral sections were incubated with the Goat Anti-Mouse IgG H&L (Alexa Fluor® 647) (ab150115, Abcam, Cambridge, UK). Pictures were taken using an Axioskop 2 immunofluorescence microscope (Carl Zeiss Microscopy GmbH, Jena, Germany) equipped with a CoolCube 1 CCD camera (MetaSystems, Newton, Massachusetts, USA).”
- Comments 12: In the present experimental model, the direct skin application of imiquimod, why were the changes in the protein expressions of tight junctions apposite in intestine and brain? They should be discussed.
Response 12: Skin application of imiquimod activates plasmacytoid dendritic cells recruited to the skin. These cells release huge amount of interferon upon their activation, which could induce the transcription of many genes (“interferon signature”). This inflammatory cascade was correlated in many studies to gut barrier dysfunction and changes of intestinal tight junctions. A translocation of bacteria will ensue promoting a systemic expansion of the inflammatory response especially interferon. This could be the pathophysiological pathway to the modification of tight junction expression in the blood-brain barrier. This was integrated in the discussion as requested as a possible explanation of our pilot study results.
- Comments 13: 3.4 and Figure 4, the order of A-D and the text would be better to be replaced to match in order from A to D. Moreover, Figure 4A and 4B, did not lupus have significant difference compared to sham?
Response 13: We changed the order as requested. As for Figure 4A and 4B we didn’t find significant difference between sham and lupus. This could be due to the number of mice in each group. Future studies will be necessary to expand these results.
Round 2
Reviewer 2 Report
Comments and Suggestions for Authors
The authors addressed the reviewer's comments and improved the manuscript.
Author Response
We sincerely appreciate and are grateful for your positive feedback and your approval.